# Growth of α-Ga_2_O_3_ from Gallium Acetylacetonate under HCl Support by Mist Chemical Vapor Deposition

**DOI:** 10.3390/nano14141221

**Published:** 2024-07-18

**Authors:** Tatsuya Yasuoka, Li Liu, Giang T. Dang, Toshiyuki Kawaharamura

**Affiliations:** 1School of Systems Engineering, Kochi University of Technology, 185 Miyanokuchi, Tosayamada, Kami 782-8502, Kochi, Japan; 256015k@gs.kochi-tech.ac.jp (T.Y.);; 2Center for Nanotechnology, Research Institute, Kochi University of Technology, 185 Miyanokuchi, Tosayamada, Kami 782-8502, Kochi, Japan

**Keywords:** mist CVD, α-Ga_2_O_3_, HCl support, surface roughness, crystallinity, growth mechanism

## Abstract

α-Ga_2_O_3_ films were grown on a *c*-plane sapphire substrate by HCl-supported mist chemical vapor deposition with multiple solution chambers, and the effect of HCl support on α-Ga_2_O_3_ film quality was investigated. The growth rate monotonically increased with increasing Ga supply rate. However, as the Ga supply rate was higher than 0.1 mmol/min, the growth rate further increased with increasing HCl supply rate. The surface roughness was improved by HCl support when the Ga supply rate was smaller than 0.07 mmol/min. The crystallinity of the α-Ga_2_O_3_ films exhibited an improvement with an increase in the film thickness, regardless of the solution preparation conditions, Ga supply rate, and HCl supply rate. These results indicate that there is a low correlation between the improvement of surface roughness and crystallinity in the α-Ga_2_O_3_ films grown under the conditions described in this paper.

## 1. Introduction

Gallium oxide (Ga_2_O_3_) is an ultra-wide bandgap material that has been extensively studied due to its attractive physical properties. It has five crystal structures: α, β, ε, δ, and γ [1,2]. Among these, β-Ga_2_O_3_, with a bandgap of 4.9 eV [3,4], can be grown by melt methods [5,6,7,8] and is expected to find use in power devices and other applications [9,10]. On the other hand, α-Ga_2_O_3_, with a corundum crystal structure, has the largest bandgap of 5.3 eV [11] (5.61 eV) [12] among the gallium oxides, and bandgap engineering between 3.7 and 8.7 eV is possible by mixing with other corundum-structured materials to form α-(M_x_Ga_1−x_)_2_O_3_ (M = Al [13,14,15,16,17], In [18,19], and Fe [20]). Although α-Ga_2_O_3_ is a thermally metastable phase, its growth on sapphire substrates was successfully demonstrated by mist chemical vapor deposition (mist CVD) first [11]. To date, α-Ga_2_O_3_ has been grown through a variety of techniques, including mist CVD [11,21,22,23,24], halide vapor phase epitaxy (HVPE) [25,26,27], metal organic chemical vapor deposition (MOCVD) [28,29], molecule beam epitaxy (MBE) [15,30,31], and atomic layer deposition (ALD) [32,33]. In their study, Uno et al. proposed that the acetylacetonate complex, which is used as a precursor in the mist CVD process, promotes the growth of α-Ga_2_O_3_ [24]. Nevertheless, the growth of α-Ga_2_O_3_ has also been reported when gallium trichloride (GaCl_3_) and gallium tribromide (GaBr_3_), which do not contain acetylacetone, were used as precursors [17,21,23]. Furthermore, the mist CVD enables the selective growth of β-, α-, ε-, δ-, and γ-Ga_2_O_3_ by modifying the substrate [34,35,36,37]. At this stage, the underlying growth mechanism of Ga_2_O_3_ by mist CVD remains unclear.

Previous reports have shown that hydrochloric acid (HCl) contributes to the phase-selective growth and quality improvement of α-Ga_2_O_3_ by HVPE [27], MOCVD [28], and mist CVD [21,22]. In those reports, it is suggested that HCl present in the reaction field affects the surface energy of the sapphire substrate due to the presence of Al-Cl bonds [27] and contributes to the stability of each crystalline phase [28]. With regard to the case of mist CVD with one solution chamber, it has been suggested that HCl in the gallium acetylacetonate (Ga(acac)_3_) precursor solution contributes to the state of Ga in solution [22]. Our research group examined the effect of HCl support by using mist CVD with multiple solution chambers on the use of GaCl_3_ as a precursor [21]. Our findings indicated that the growth rate of α-Ga_2_O_3_ increased with increasing HCl supply, and that the larger Ga supply rate and HCl support suppressed the incorporation of carbon impurities into the film, resulting in improved surface roughness. However, the effect of HCl on Ga_2_O_3_ growth and the underlying mechanisms remain incompletely understood and necessitate further investigation. Thus, this study aims to investigate the effect of HCl support on α-Ga_2_O_3_ growth by mist CVD with multiple solution chambers when Ga(acac)_3_ is employed as a precursor. It was found that HCl support resulted in a discrepancy in the trends of growth rate and surface roughness variation in comparison to the case when GaCl_3_ is employed as a precursor. In addition, the effect of two starting materials, Ga(acac)_3_ and GaCl_3_, on the crystallinity of α-Ga_2_O_3_ was also examined.

## 2. Experimental Methods

α-Ga_2_O_3_ thin films were grown on *c*-plane sapphire substrates using a mist CVD system, which is described in Figure 1 [38]. The growth conditions are summarized in Table 1. Ga(acac)_3_ was dissolved in a mixture of deionized water and hydrochloric acid (HCl), and gallium trichloride, GaCl_3_, was dissolved in deionized water. Each Ga solution was stirred until completely dissolved for a minimum of one hour. These solutions were used, respectively, as Ga precursor solutions. The Ga(acac)_3_ solution was set at concentrations of 20–100 mM and GaCl_3_ was set at concentrations of 50–300 mM. HCl diluted in de-ionized water at concentrations of 0.57–1.13 M was used as a support to improve the quality of α-Ga_2_O_3_. Each solution was individually atomized using ultrasonics and then supplied to the fine-channel reactor with carrier and dilution gases. If multiple solutions were used, they were supplied to the reactor via the mixing chamber. Ga and HCl supply rates were controlled by solution concentrations and flow rates of carrier gas. Mist CVD with multiple solution chambers, also known as third-generation (3rd G) mist CVD, allows mist generated from multiple solutions to be supplied to the reaction field without contributing to the solution state. This process suppresses side reactions and makes it suitable for composition control [13], doping [39], and the introduction of reaction support agents [21]. Details of this mist CVD system were described in Ref. [38]. The growth temperature was set at 400 °C.

The crystal structure and crystallinity were analyzed using 2θ/ω scan X-ray diffraction (XRD) and X-ray rocking curves, respectively (SmartLab, Rigaku, Tokyo, Japan). The thickness of α-Ga_2_O_3_ films on single-side mirror sapphire substrates was measured by spectroscopic ellipsometry (M-2000DI, J.A. Woollam, Lincoln, U.S.A.) and calculated from Laue fringe spacing. Atomic force microscopy (AFM, Cypher, Asylum Research, Santa Barbara, U.S.A.) was used to analyze the surface roughness.

## 3. Results and Discussions

### 3.1. α-Ga_2_O_3_ Film Thicknesses Estimated from Ellipsometry and from Laue Fringes in XRD Spectra

Figure 2a shows a typical XRD spectrum of the α-Ga_2_O_3_ film grown in this study. When the film thickness was less than about 250 nm, Laue fringes were clearly observed around the α-Ga_2_O_3_ peak, as shown in Figure 2a. Figure 2b shows the α-Ga_2_O_3_ film thickness estimated from the spacing of the Laue fringes in the XRD spectra and ellipsometry. The estimated film thickness errors were less than 3% and 1%, respectively. The α-Ga_2_O_3_ films were grown on single-sided mirror *c*-plane sapphire substrates with 50–100 mM Ga(acac)_3_ precursor solution in this study. The dashed line in Figure 2b indicates that the values on the vertical and horizontal axes are the same. Figure 2b shows similar thicknesses, as estimated from the spacing of the Laue fringes in the XRD spectra and from ellipsometry. A comparison between thicknesses determined from Laue spacings and measured by direct methods such as profilers [40]/cross-sectional transmission electron microscopy (TEM) images [41] also shows a discrepancy of <5%. This suggests that for α-Ga_2_O_3_ grown on sapphire substrates, the film thickness calculated by Laue fringe spacing is close to the actual film thickness. Therefore, the thickness of α-Ga_2_O_3_ films on sapphire substrates was determined based on the spacings of Laue fringes in this study.

### 3.2. Effect of HCl on α-Ga_2_O_3_ Growth Using Ga(acac)_3_ as a Starting Material

Figure 3a,b show the α-Ga_2_O_3_ growth rate versus Ga supply rate and HCl supply rate when Ga(acac)_3_ is used as the Ga precursor. The colors indicate the concentrations of the Ga solutions, while solid dots (●) and crosses (×) indicate data points of the α-Ga_2_O_3_ growths with and without HCl support, respectively. It should be noted that HCl is added in the solvent during Ga solution preparation to dissolve Ga(acac)_3_ and is also supplied additionally during growth from a separated HCl solution. Consequently, the HCl supply rate represents the total amount of HCl (i) contained in Ga solutions and (ii) supplied from the dedicated HCl source. The HCl supply rate is not zero when HCl is not supplied and the data points are plotted as crosses (×) in this case. Figure 3a clearly shows that the growth rate increased monotonically with increasing Ga supply rate. At the Ga concentration of 20 mM, the growth rates with and without HCl support were similar. However, at the Ga concentrations higher than 20 mM, the growth rates without HCl support were smaller than that with HCl support. This difference in the growth rates may be attributed to the formation of oligomers (or supermolecules) at higher Ga concentrations without HCl support, which prevented Ga molecules from reaching the substrate due to steric hindrance, as illustrated in Figure 4. Conceivably, it is expected that HCl will inhibit the formation of oligomers. In Figure 3b, it was observed that the growth rate remained constant with increasing HCl supply rate at a Ga solution concentration of 20 mM. In contrast, the growth rate seems to increase slightly with increasing HCl supply rate, despite the Ga supply rate being almost constant at Ga solution concentrations of 80 and 100 mM. This trend was similar to that observed when GaCl_3_ was used as a precursor [21].

Figure 5a,b show the results of plotting the TPR (thickness per surface roughness), which is the film thickness divided by the RMS surface roughness, against the Ga and HCl supply rates. It should be noted that the TPR is capable of distinguishing between 2D and 3D growth modes and can be employed in the evaluation of surface roughness of samples with different film thicknesses [21]. As shown in Figure 5a, the TPR increases with decreasing Ga supply rates when the Ga supply rate is below approximately 0.07 mmol/min. Conversely, when the Ga supply rate exceeds ~0.07 mmol/min, the TPR remains constant at approximately 30. Figure 5c,d show the AFM images of samples A and B, which were marked by arrows in Figure 5a and Figure 5b, respectively. The values of *T_L_* and RMS shown in the top and bottom of the AFM images present the thickness and RMS surface roughness, respectively. Although samples A and B exhibit nearly identical film thicknesses, the TPR value is larger for sample A due to its smaller surface roughness. However, the AFM images demonstrate that both A and B exhibit island growth. The surface morphology of all samples grown in this study was similar to that of samples A and B. When the TPR was small, the surface was composed of small surface grains, as observed in sample B (Figure 5d). Conversely, when the TPR increases, the surface grain size increases, as observed in sample A (Figure 5c). While sample A was, in fact, a 3D growth, the trend of surface roughness change with film thickness is analogous to that of 2D growth, indicating that it is a 2D-like growth mode. Therefore, when growing α-Ga_2_O_3_ using Ga(acac)_3_, the growth mode is 3D when the TPR is about 30 and 2D-like when it is larger. According to a previous report, the film grows in a 2D-like growth mode when the impurity concentration in the film is reduced by HCl support [21]. It is assumed that the carbon impurity concentration in the film is reduced by HCl support through a similar mechanism, resulting in a 2D-like growth mode when Ga(acac)_3_ is used as the starting material. However, Ga(acac)_3_ has 15 C atoms for each Ga atom, unlike GaCl_3_. Therefore, it is expected that HCl-supported growth did not improve the surface roughness of the films when the Ga supply rate became large enough to suppress impurity uptake in the films.

### 3.3. Growth Characteristics of α-Ga_2_O_3_ for HCl Support in Each Starting Material

Figure 6 shows a summary of the effect of HCl support on α-Ga_2_O_3_ growth when Ga(acac)_3_ and GaCl_3_ are employed as starting materials. Note that the experimental data of the GaCl_3_ results are presented in Ref. [21]. When Ga(acac)_3_ was employed as the starting material, the growth rate exhibited a monotonic increase with increasing Ga supply rate. However, when the Ga supply rate exceeded 0.1 mmol/min, the growth rate demonstrated an additional increase with increasing HCl supply rate, in addition to the Ga supply rate. In contrast, when GaCl_3_ was employed, the growth rate exhibited an increase solely in response to an increasing HCl supply rate. With regard to surface roughness, it was observed that there was no dependence on the Ga and HCl supply rates when Ga(acac)_3_ was employed as the starting material, provided that the Ga supply rate was larger than 0.07 mmol/min. However, when the Ga supply rate was smaller, the surface roughness was improved by HCl support. In contrast, when GaCl_3_ was employed, there was no correlation between the Ga and HCl supply rates when the Ga supply rate was less than 0.1 mmol/min. However, the surface roughness exhibited an improvement with an increase in the Ga and HCl supply rates when the Ga supply rate was higher.

Figure 7a,b illustrate the outcomes of plotting the thickness dependence of the full width at half maximum (FWHM) of the α-Ga_2_O_3_ (0006) peak grown on sapphire, obtained through X-ray rocking curve measurements on linear and logarithmic axes, respectively. In Figure 7a,b, the thickness of the Ga_2_O_3_ film on a quartz substrate, which was grown simultaneously with the film on sapphire, was measured by ellipsometry and utilized since the fringes are not visible when the film is thick. The FWHM decreased inversely with increasing film thickness under all growth conditions, regardless of solute species, Ga supply rate, or HCl supply rate. This indicates that the crystallinity of α-Ga_2_O_3_ is not improved by the solution preparation conditions or HCl support when grown by mist CVD. In addition, when plotted on a logarithmic axis, as shown in Figure 7b, there is a slight difference depending on the starting material, fitting y = 5503.6x ^− 0.795^ and y = 2080.9x ^− 0.592^ when Ga(acac)_3_ and GaCl_3_ are used as the starting materials, respectively. At this stage, there is no adequate physical explanation as to why differences in starting materials cause differences in the thickness dependence of crystallinity. This may be attributed to the amount of impurities, such as carbon and chloride atoms, incorporated into the film as impurities. It is anticipated that future growth of α-Ga_2_O_3_ using alternative starting materials, such as GaBr_3_, gallium nitrate (Ga(NO_3_)_3_), and gallium triiodide (GaI_3_), will elucidate the underlying cause of this trend. In any case, it can be concluded that there is a low correlation between the improvement of surface roughness and crystallinity in the α-Ga_2_O_3_ film grown under the conditions described in this paper.

### 3.4. Facts Revealed in α-Ga_2_O_3_ Grown by Mist CVD

Table 2 presents a summary of the controllability of surface roughness and crystallinity in α-Ga_2_O_3_ films. With regard to surface roughness, it was found that this property could be controlled by the type of precursor, Ga supply rate, and HCl support. In contrast, crystallinity was found to improve only with increasing film thickness, regardless of the Ga supply rate and HCl support, although there were some differences in the results depending on the type of starting material.

## 4. Conclusions

This study investigated the effect of HCl support on film quality during the growth of α-Ga_2_O_3_ by mist CVD using Ga(acac)_3_ as a precursor. It also examined the difference in crystallinity resulting from different starting materials. The growth rate monotonically increased with increasing Ga supply rate. However, as the Ga solution concentration increased, the growth rate further increased with increasing HCl supply rate, showing the same trend as that of GaCl_3_. When the Ga solution concentration exceeded 20 mM, the growth rate without HCl support was smaller than that with HCl support. This effect is believed to be caused by the formation of oligomers at high Ga solution concentrations, which inhibited Ga molecules from reaching the substrate due to steric hindrance. It is therefore anticipated that HCl support will be effective in inhibiting the formation of oligomers. Furthermore, the HCl support resulted in improved surface roughness of α-Ga_2_O_3_ due to the suppression of carbon impurities incorporated into the film, similarly to GaCl_3_. However, there was a difference in the amount of Ga supply between Ga(acac)_3_ and GaCl_3_ when they were employed as starting materials, respectively. In the case of Ga(acac)_3_, surface roughness tended to improve with smaller amounts of Ga supply. Ga(acac)_3_ has 15 C atoms for each Ga atom, unlike GaCl_3_. Therefore, it is expected that with high Ga supply rates, the HCl support is not sufficient to suppress the incorporation of impurities into the films, leading to no improvement in surface roughness. Crystallinity improved with increasing film thickness, regardless of the solution preparation conditions, including solute species and concentration, Ga supply rate, and HCl supply rate. This indicates that HCl support during α-Ga_2_O_3_ crystal growth by mist CVD has a negligible effect on crystallinity improvement. However, the power-approximation results varied slightly due to differences in starting materials. In addition, the findings indicate a low correlation between the improvement of surface roughness and crystallinity in the α-Ga_2_O_3_ films grown under the conditions described in this paper.

## Figures and Tables

**Figure 1 nanomaterials-14-01221-f001:**
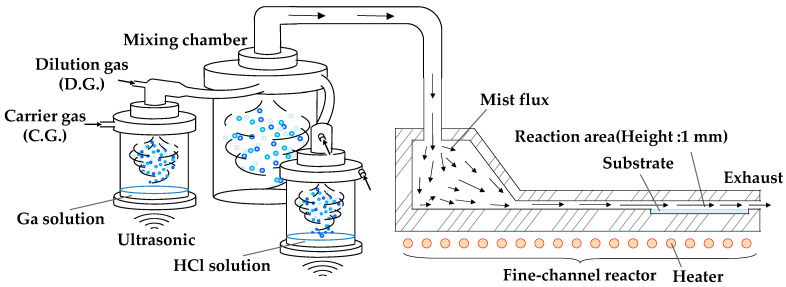
A schematic image of mist CVD system with multiple solution chambers, a mixing chamber, and a fine-channel reactor [38].

**Figure 2 nanomaterials-14-01221-f002:**
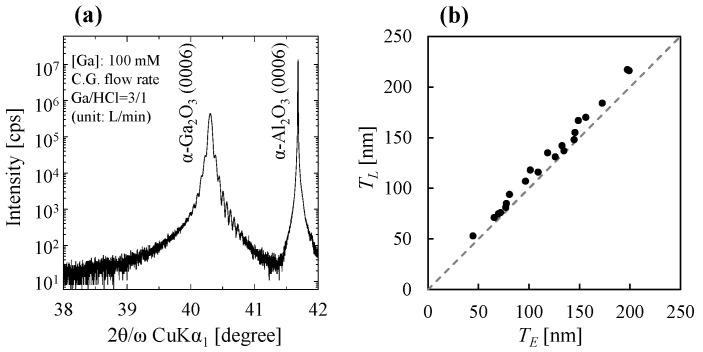
(**a**) Typical XRD spectra of α-Ga_2_O_3_ film grown on *c*-plane sapphire substrate and (**b**) thickness estimation results using different methods (vertical axis *T_L_*: thickness calculated from the spacings of Laue fringes in XRD spectra; horizontal axis *T_E_*: thickness calculated from spectroscopic ellipsometry). The dashed line represents the same value on both the vertical and horizontal axes.

**Figure 3 nanomaterials-14-01221-f003:**
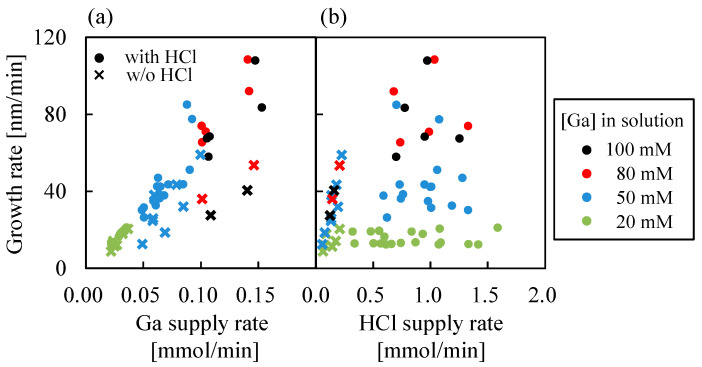
Dependence of the α-Ga_2_O_3_ growth rates on (**a**) Ga supply rate and (**b**) HCl supply rate.

**Figure 4 nanomaterials-14-01221-f004:**
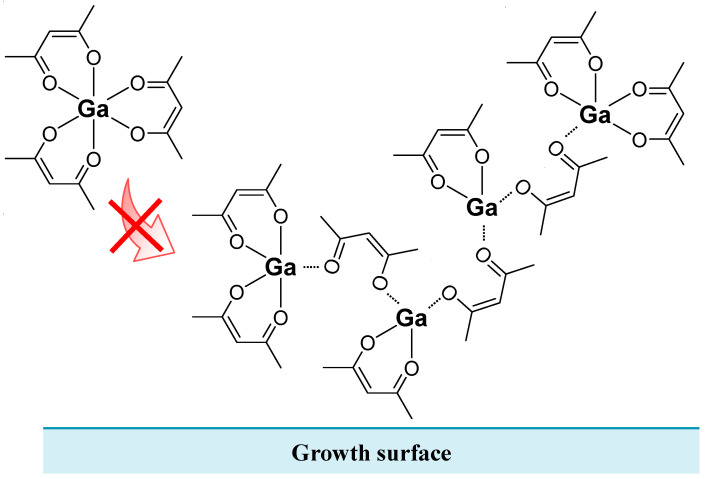
A schematic image of an oligomer acting as a steric hindrance, preventing Ga molecules from attaching to the growth surface.

**Figure 5 nanomaterials-14-01221-f005:**
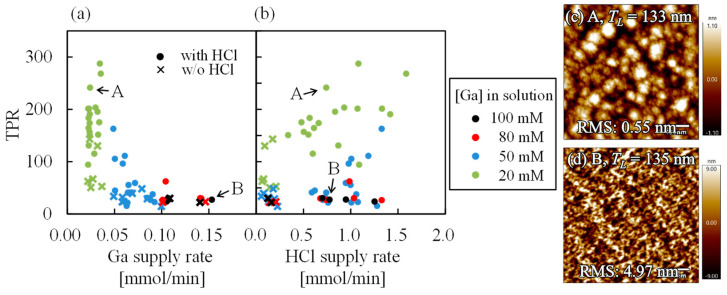
Dependence of the TPR on (**a**) Ga supply rate and (**b**) HCl supply rate. (**c**,**d**) show the AFM images of samples A and B, respectively.

**Figure 6 nanomaterials-14-01221-f006:**
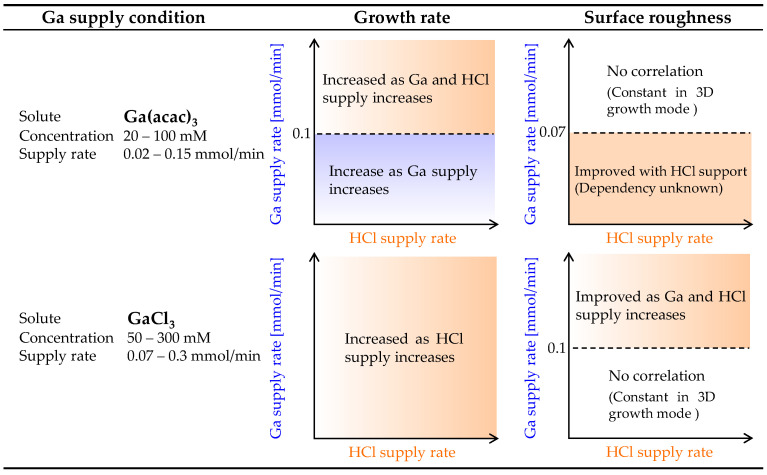
A summary of the effect of HCl support on the growth rate and surface roughness of α-Ga_2_O_3_ thin films when Ga(acac)_3_ and GaCl_3_ are employed as starting materials.

**Figure 7 nanomaterials-14-01221-f007:**
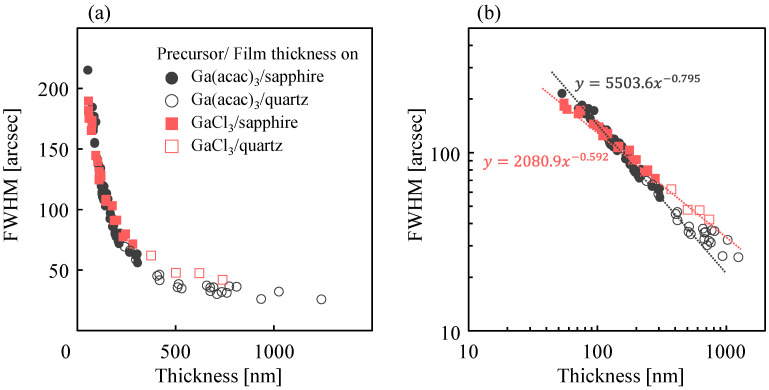
The results of plotting the thickness dependence of the full width at half maximum (FWHM) of the α-Ga_2_O_3_ (0006) peak on *c*-plane sapphire substrate measured by X-ray rocking curve on (**a**) linear and (**b**) logarithmic axes.

**Table 1 nanomaterials-14-01221-t001:** Summary of growth conditions of α-Ga_2_O_3_ thin films.

Solution	Ga Solution		HCl Solution
Solute	Ga(acac)_3_ ^(a)^	GaCl_3_ ^(b)^	HCl ^(c)^
Concentration	20–100 mM	50–300 mM	0.57–1.13 M
Solvent (mixing ratio)	DI water, HCl (199:1, 99:1)	DI water	DI water ^(d)^
Carrier and dilution gas	N_2_		
Substrate	*c*-plane sapphire		
Growth temperature	400 °C		
Ultrasonic transducer	2.4 MHz, 24 V–0.625 A (×3)		

^(a)^ Gallium acetylacetonate, 99.99%, Sigma-Aldrich, St. Louis, U.S.A. ^(b)^ Gallium trichloride, solution with 30 wt%, Kojundo Chemical Laboratory, Saitama, Japan. ^(c)^ Hydrochloric acid, 35–37%, Wako Pure Chemical Corporation, Osaka, Japan. ^(d)^ De-ionized water, Merch Millipore, Burlington, U.S.A.

**Table 2 nanomaterials-14-01221-t002:** Controllability of surface roughness and crystallinity in α-Ga_2_O_3_ films.

	Precursor	Ga Supply Rate	HCl Support
Surface roughness	✓	✓	✓
Crystallinity	✓	-	-

## Data Availability

Data are contained within the article.

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
