# Peer review of "Growth of α-Ga2O3 from Gallium Acetylacetonate under HCl Support by Mist Chemical Vapor Deposition"

_nanomaterials, 2024, doi:10.3390/nano14141221_

Round 1

Reviewer 1 Report

Comments and Suggestions for Authors

In this paper, α-Ga2O3 films were prepared by common HCl-supported mist CVD. The authors investigated the effect of HCl-support on α-Ga2O3 film quality and revealed a low correlation between the improvement of surface roughness and crystallinity in the α-Ga2O3 films. Although the paper is well organized, some crucial issues need to be modified before further consideration.

(1) In Figure 2(b), the authors claimed that there was no discernible trend in the growth rate depending on the HCl supply rate. They observe the growth rate remained constant with increasing HCl supply rate at a Ga solution concentration of 20 mM. But the following description “the growth rate seems to increase slightly with increasing HCl supply rate…” contradicts it.

(2) The atomic force microscopy (AFM) images of α-Ga2O3 films prepared at different conditions, such as precursor, Ga supply rate, HCl support should be supplied for intuitively comparing their surface roughnesses.

(3) The self-citations are more than 8 in this paper.

Author Response

Comment1: In Figure 2(b), the authors claimed that there was no discernible trend in the growth rate depending on the HCl supply rate. They observe the growth rate remained constant with increasing HCl supply rate at a Ga solution concentration of 20 mM. But the following description “the growth rate seems to increase slightly with increasing HCl supply rate…” contradicts it.

Response1: Thank you for your comment. We apologize for the confusion caused by the description of the complicated data. What we meant was that there was no discernible general trend across all Ga concentrations. For individual Ga concentrations we observed that at 20 mM, the growth rate remained constant, whereas at 80 mM and 100 mM the growth rate appeared to increase with increasing HCl supply.

Changes in the manuscript:

The confusing sentences: “Figure 2(b) indicates that there is no discernible trend in the growth rate depending on the HCl supply rate. However, when focusing on the tendency of growth rates with changes in each Ga solution concentration,” were removed. We kept only main information that we want to transmit: “In Fig. 3(b), it was observed that the growth rate remained constant with increasing HCl supply rate at a Ga solution concentration of 20 mM. In contrast, the growth rate seems to increase slightly with increasing HCl supply rate, despite the Ga supply rate being almost constant at Ga solution concentrations of 80 and 100 mM.” (Please, note that the figure number has been changed to 3, because another figure has been added to accommodate other referees’ comments.)  (p4, lines 145 – 149)

Comment2: The atomic force microscopy (AFM) images of α-Ga2O3 films prepared at different conditions, such as precursor, Ga supply rate, HCl support should be supplied for intuitively comparing their surface roughnesses.

Response2: Thank you for the recommendation. Two representative AFM images have been added (Figure 5(c) and (d)). Both samples have the same amount of HCl support, but different Ga solution concentrations, which result in significant differences in TPR values. The surface morphologies of the Ga2O3 produced in this study were all 3D grown, as illustrated in Fig. 5(c) and (d). However, as the TPR increased, the surface grain size increased, as shown in Figure 5(c) compared to Figure 5(d). While α-Ga2O3 with high TPR value was, in fact, a 3D growth, the trend of surface roughness change with film thickness is analogous to that of 2D growth, indicating that it is a 2D-like growth mode. We have added these details to the text. (p5, lines 167 - 178)

Comment3: The self-citations are more than 8 in this paper.

Response3: We have addressed the issue of self-citation and have reduced the number of self-cited references to less than 15% of the total references.

Reviewer 2 Report

Comments and Suggestions for Authors

This paper investigates Growth of α-Ga2O3 from Gallium Acetylacetonate under HCl. The topic of the work is interesting but should be significantly expanded. The work should supported more experimental data. Some comments are listed as follows:

1) Please explain the abbreviations used in the text such as CVD, HVPE, etc.,

2) Please show AFM microscope images

3) Also X-ray diffraction profiles of a-Ga2O3 should be provided. 

4) Please show the difference between the layers in more detail.

5) Measurement errors should also be discussed.

Author Response

This paper investigates Growth of α-Ga2O3 from Gallium Acetylacetonate under HCl. The topic of the work is interesting but should be significantly expanded. The work should supported more experimental data. Some comments are listed as follows:

Comment1: Please explain the abbreviations used in the text such as CVD, HVPE, etc.,

Response1: Thank you for pointing this out. We have spelled out the abbreviations.

Comment2/4: Please show AFM microscope images. / Please show the difference between the layers in more detail.

Response2/4: Thank you for the recommendation. Two representative AFM images have been added (Figures 5(c) and (d)). Both samples have the same amount of HCl support, but different Ga solution concentrations, which result in significant differences in TPR values. The surface morphologies of the Ga2O3 produced in this study were all 3D grown, as illustrated in Figs. 5(c) and (d). However, as the TPR increased, the surface grain size increased, as shown in Figure 5(c) compared to Figure 5(d). While α-Ga2O3 with high TPR value was, in fact, a 3D growth, the trend of surface roughness change with film thickness is analogous to that of 2D growth, indicating that it is a 2D-like growth mode. We have added these details to the text. (p5, lines 167 - 178)

Comment3: Also X-ray diffraction profiles of a-Ga2O3 should be provided. 

Response3: A representative XRD spectrum was added: Laue fringes were observed around α-Ga2O3 (0006) peak as shown in Fig. 2(a), which was used to estimate the thickness of the α-Ga2O3 Figure 2(b) plots the film thickness estimated by ellipsometry and Laue fringes for α-Ga2O3 grown on a single-sided mirror sapphire substrate under several conditions. This demonstrates that the estimation of film thickness by Laue fringes is valid. (p3, line 101 – p4, line 125)

Comment5: Measurement errors should also be discussed.

Response5:We have performed an error analysis and found that the relative errors for the thicknesses determined from the Laue fringes are < 5%, as shown in the table below.

For Laue fringes:

Samples

By Laue fringes

By direct methods

Discrepancy

Reference

nm

nm

a-Ga2O3 on sapphire

215

210±10a

2.3%

Dang et al, IEEETED 62, 3640 (2015)

a-Ga2O3 on a patterned buffer

482

491±13b

1.9%

Dang et al, APL 119, 041902 (2021)

aDetermined by a profiler after etching

bDetermined from a cross-sectional Transmission Electron Microscope image

Changes in the manuscript:

We have added Figs. 2(a) and (b) in the revised manuscript and discussed about the errors accordingly (pages 3 and 4, lines 101-125)

Reviewer 3 Report

Comments and Suggestions for Authors

In this work, the authors investigated the Growth of α-Ga2O3 from Gallium Acetylacetonate under HCl Support by Mist Chemical Vapor Deposition. This study examined the impact of HCl support on film quality during α-Ga2O3 growth via mist Chemical Vapor Deposition. The results are well discussed, and the relationship between chemical composition, crystal structure, and physical properties is well established.

However, some imperfections remain, and I ask the authors to rectify them.

1.      The discussion in the introductory part seems to be able to sort it out further. For example: Should the brief discussion is the Growth of α-Ga2O3 from Gallium Acetylacetonate under HCl or other materials Support by Mist Chemical Vapor Deposition.

2.      In the sentence " The crystal structure was analyzed using the 2θ/ω scan X-ray diffraction (XRD, 83 Rigaku, SmartLab). " The author should give a detailed explanation and it is necessary to include XRD diffractograms.

3.      It is necessary to specify the origin of figure 1.

4.      Some highly relevant studies must be added.

5.      The novelty of the study is low and should be improved.

6.      The overall grammar and language should be improved.

7.      In the sentence " Figure 5 shows a summary of the effect of HCl support on α-Ga2O3 growth when Ga(acac)3 and GaCl3 are employed as starting materials. Note that the details of the GaCl3 results are presented in Ref. 23. " The author should give a detailed explanation.

Comments on the Quality of English Language

The overall grammar and language should be improved.

Author Response

In this work, the authors investigated the Growth of α-Ga2O3 from Gallium Acetylacetonate under HCl Support by Mist Chemical Vapor Deposition. This study examined the impact of HCl support on film quality during α-Ga2O3 growth via mist Chemical Vapor Deposition. The results are well discussed, and the relationship between chemical composition, crystal structure, and physical properties is well established.

However, some imperfections remain, and I ask the authors to rectify them.

Comment1: The discussion in the introductory part seems to be able to sort it out further. For example: Should the brief discussion is the Growth of α-Ga2O3 from Gallium Acetylacetonate under HCl or other materials Support by Mist Chemical Vapor Deposition.

Response1: Thank you for the suggestion. In the introduction, we have added the discussion on the growth of α-Ga2O3from gallium acetylacetonate and gallium trichloride (GaCl3). We couldn’t find any works that use other supporting materials than HCl to enhance the quality of α-Ga2O3 in the mist CVD process. It has been reported that acetylacetonate complexes facilitate α-Ga2O3 In this work, they used acetylacetone as a means of elucidating the deposition mechanism.

Changes in the manuscript: We have incorporated the information into the text. (page1, lines 35-42)

Comment2: In the sentence " The crystal structure was analyzed using the 2θ/ω scan X-ray diffraction (XRD, 83 Rigaku, SmartLab). " The author should give a detailed explanation and it is necessary to include XRD diffractograms.

Response2: Thank you for the recommendation. A representative XRD spectrum was added: Laue fringes were observed around α-Ga2O3 (0006) peak as shown in Fig. 2(a), which was used to estimate the thickness of the α-Ga2O3 Figure 2(b) plots the film thickness estimated by ellipsometry and Laue fringes for α-Ga2O3 grown on a single-sided mirror sapphire substrate under several conditions. This demonstrates that the estimation of film thickness by Laue fringes is valid. (p3, line 101 – p4, line 125)

Comment3: It is necessary to specify the origin of figure 1.

Response3: This is an original figure, but we have already published a similar figure from our research group, so we have added a reference to it.

Comment4/5: Some highly relevant studies must be added. / The novelty of the study is low and should be improved.

Response4/5: The previous study revealed that the acetylacetonate complex promotes Gaâ‚‚O₃ growth in the mist CVD method and provided insights into the growth mechanism. However, the growth of Gaâ‚‚O₃ has been reported even when GaCl₃ and GaBr3 were used. In addition, mist CVD enables the selective growth of β-, α-, ε-, δ-, and γ-Ga2O3 by modifying the substrate, and the growth mechanism of Ga2O3 by mist CVD remains unclear. The findings of this study may prove valuable in gaining insight into the growth mechanism of Ga2O3. We have incorporated this information into the report. (page1, lines 35-42)

Comment6: The overall grammar and language should be improved.

Response6: Due to the time constraint, we weren’t able to have the manuscript proofread by native speakers. However, we have carefully read the manuscripts, eliminated mistakes, and run the grammar and usage check by popular software.

Comment7: In the sentence " Figure 5 shows a summary of the effect of HCl support on α-Ga2O3 growth when Ga(acac)3 and GaCl3 are employed as starting materials. Note that the details of the GaCl3 results are presented in Ref. 23. " The author should give a detailed explanation.

Response7: The text in the manuscript provides details of the results, but does not include the actual experimental data. The text has been revised. (p6, line 195, “Note that the details” -> “Note that the experimental data”).

Reviewer 4 Report

Comments and Suggestions for Authors

Manuscript entitled "Growth of α-Ga2O3 from Gallium Acetylacetonate under HCl Support by Mist Chemical Vapor Deposition" is devoted to study the effect of the HCl addition on the parameters of a-Ga2O3 deposited by mist-CVD method using Ga-containing precursors. The patterns of changes in the growth rate and surface roughness of gallium oxide films depending on the growth parameters have been determined. The article is certainly of interest from the point of view of establishing the patterns of growth of Ga2O3 films using the mist-CVD. However, the article in its current edition cannot be published in the journal "Nanomaterials" for the following reasons.

1. In the article, one of the main issues discussed is the growth mechanism (2D or 3D) of films when the production conditions change. At the same time, no evidence is given for these factors, although this can be demonstrated by atomic force or scanning electron microscopy.

2. Determination of film thickness from oscillations in XRD spectra is incorrect in the case of 3D growth, which leads to an error in determining the growth rate in the case of such mechanism.

3. The X-ray diffraction spectra of the samples are not shown, which reduces the information content of the presented results and does not allow one to clearly assess the adequacy of the applied method for determining the film thickness.

4. The meaning of Figure 6b is completely unclear. If any pattern is obtained as a result of fitting, it should be explained from the point of view of the physical laws underlying it, and comments should be given regarding the values ​​of the fitting coefficients.

5. The main text of the article is a listing of the results obtained without any detailed discussion. The conclusion also lists only the results obtained.

Nevertheless, this manuscript contains some new interesting results and could be reviewed again after major revisions.

Author Response

Manuscript entitled "Growth of α-Ga2O3 from Gallium Acetylacetonate under HCl Support by Mist Chemical Vapor Deposition" is devoted to study the effect of the HCl addition on the parameters of a-Ga2O3 deposited by mist-CVD method using Ga-containing precursors. The patterns of changes in the growth rate and surface roughness of gallium oxide films depending on the growth parameters have been determined. The article is certainly of interest from the point of view of establishing the patterns of growth of Ga2O3 films using the mist-CVD. However, the article in its current edition cannot be published in the journal "Nanomaterials" for the following reasons.

Comment1: In the article, one of the main issues discussed is the growth mechanism (2D or 3D) of films when the production conditions change. At the same time, no evidence is given for these factors, although this can be demonstrated by atomic force or scanning electron microscopy.

Response1: Thank you for the critical comment. Two representative AFM images have been added (Figures 5(c) and (d)). Both samples have the same amount of HCl support, but different Ga solution concentrations, which result in significant differences in TPR values. The surface morphologies of the α-Ga2O3 produced in this study were all 3D grown, as illustrated in Figs. 5(c) and (d). However, as the TPR increased, the surface grain size increased, as shown in Figure 5(c) compared to Figure 5(d). While α-Ga2O3 with high TPR value was, in fact, a 3D growth, the trend of surface roughness change with film thickness is analogous to that of 2D growth, indicating that it is a 2D-like growth mode. We have added these details to the text. (p5, lines 167 - 178)

Comment2/3: Determination of film thickness from oscillations in XRD spectra is incorrect in the case of 3D growth, which leads to an error in determining the growth rate in the case of such mechanism. / The X-ray diffraction spectra of the samples are not shown, which reduces the information content of the presented results and does not allow one to clearly assess the adequacy of the applied method for determining the film thickness.

Response2/3: Thank you for the critical comment. We believe that the Laue method provide thicknesses within 5% of actual thicknesses for films of up to ~ 480 nm, as shown in the table below:

Samples

By Laue fringes

By direct methods

Discrepancy

Reference

nm

nm

a-Ga2O3 on sapphire

215

210±10a

2.3%

Dang et al, IEEETED 62, 3640 (2015)

a-Ga2O3 on a patterned buffer

482

491±13b

1.9%

Dang et al, APL 119, 041902 (2021)

aDetermined by a profiler after etching

bDetermined from a cross-sectional Transmission Electron Microscope image

A representative XRD spectrum was also added: Laue fringes were observed around α-Ga2O3 (0006) peak as shown in Fig. 2(a), which was used to estimate the thickness of the α-Ga2O3 films. Figure 2(b) plots the film thickness estimated by ellipsometry and Laue fringes for α-Ga2O3 grown on a single-sided mirror sapphire substrate under several conditions. This demonstrates that the estimation of film thickness by Laue fringes is valid. We have added this point. (p3, line 101 – p4, line 125)

Comment4: The meaning of Figure 6b is completely unclear. If any pattern is obtained as a result of fitting, it should be explained from the point of view of the physical laws underlying it, and comments should be given regarding the values ​​of the fitting coefficients.

Response4: At this stage, we don’t have an adequate physical explanation as to why differences in starting materials cause differences in the thickness dependence of crystallinity. It is anticipated that future growth of α-Ga2O3 using alternative starting materials may elucidate the cause. We have added this point in the text. (p7, lines 226 - 231)

Comment5: The main text of the article is a listing of the results obtained without any detailed discussion. The conclusion also lists only the results obtained.

Response5: As you correctly point out, the comparison of the effects of HCl support for each starting material on α-Ga2O3 growth is merely a matter of results, without any discussion of the details. However, the section on the growth of Ga(acac)3 does discuss this in more detail. The discussion on the formation of oligomers that prevent Ga molecules from reaching the growth surface is, in our opinion, valuable. For clarity, a section has been added to the text. We have added the discussion of the summary as well. (p8, lines 255 - 263) 

Round 2

Reviewer 1 Report

Comments and Suggestions for Authors

The authors have replied all of the comments and supplemented the neccessary data. I recommand its publication as it is now.

Reviewer 2 Report

Comments and Suggestions for Authors

The authors improved the work in accordance with my comments. The work may be accepted for printing.

Reviewer 4 Report

Comments and Suggestions for Authors

The manuscript could be published in the present form